# Dementia and Dependency vs. Proxy Indicators of the Active Ageing Index in Indonesia

**DOI:** 10.3390/ijerph18168235

**Published:** 2021-08-04

**Authors:** Eef Hogervorst, Elisabeth Schröder-Butterfill, Yvonne Suzy Handajani, Philip Kreager, Tri Budi W. Rahardjo

**Affiliations:** 1School of Sports, Exercise &Health Sciences, National Centre for Sports and Exercise Medicine, Loughborough University, Loughborough LE11 3TU, UK; 2Ageing and Gerontology Group, Department Economic, Social and Political Science, University of Southampton, Southampton SO17 1BJ, UK; E.Schroeder-Butterfill@soton.ac.uk; 3Department of Public Health, Faculty of Medicine, Atma Jaya Catholic University of Indonesia, Daerah Khusus Ibukota Jakarta 12930, Indonesia; yvonne.hand@atmajaya.ac.id; 4Institute of Human Sciences, University of Oxford, Oxford OX1 2JD, UK; philip.kreager@some.ox.ac.uk (P.K.); tri.budi.wr@gmail.com (T.B.W.R.); 5Faculty of Health Sciences, University of Respati, Daerah Istimewa Yogyakarta 55281, Indonesia

**Keywords:** dementia, activities of daily living, ageing well, active ageing index

## Abstract

Dementia prevalence is increasing worldwide and developing countries are expected to carry the highest burden of this. Dementia has high care needs and no current effective long-term treatment. However, factors associated with active ageing (e.g., longer employment; participation in society; independent, healthy and secure living; and enabling environments to allow people to remain psychosocially and physically active) could help maintain independence in older people for longer. We investigated proxy indicators of the Active Ageing Index (AAI), which were offset against dementia and dependency (assessed by Instrumental Activities of Daily Living or IADL) in multi-ethnic urban (Jakarta) and rural (Sumedang and Borobudur) health care districts on Java, Indonesia. Dementia was assessed using validated cognitive dementia screening tests, the IADL and carer reports. Dementia and dependency prevalence showed large interregional differences and were highest in rural Borobudur. Dementia and dependency were associated with an older age, lower education (for dementia), worse physical health (for dependency) and not engaging in psychosocial activities, such as attending community events, reading (for dementia) and sport activities (for dependency). By supporting active ageing activities in Puskesmas (primary health care centers) and improving access to medical care, rural areas could possibly reduce dementia and dependency risk. Our follow-up study planned in 2021 should illustrate whether recent relevant policies have rendered success in these areas. Using active ageing indicators could focus policies to support regions with targeted interventions to compress care needs in older people.

## 1. Introduction

Dementia is a growing health and economic concern worldwide, with absolute growth of its numbers estimated to be highest in East Asia [1]. In terms of numbers, Indonesia is in the top 5 of aging populations worldwide [2] and age is a major risk factor for dementia [3]. In 2019, ‘lansia’ or older people over 60 years of age were estimated to make up between 10% and 12% of the Indonesian population [2,4]. With this proportion of Indonesian older people growing to 21% of the population by 2050 [2], the World Alzheimer Report estimated that from around 1 million people with dementia in Indonesia in 2015, this would increase to 2 million people afflicted by 2030, which number would double again two decades later [1]. We had earlier performed dementia growth projections with similar outcomes from a survey conducted in 2006/2007 in 705 older people over 60 years of age from rural and urban sites on Java using algorithms based on validated translated and adapted cognitive tests, standard medical examinations and questionnaires [5]. This study showed large differences in dementia prevalence in those over 60 years of age between urban (3%) and rural sites (7–16%), with the highest rate in rural areas close to Yogyakarta, which also had the oldest and least educated population [5]. A recent paper [6] suggested even higher rates of dementia prevalence in this area, of 20.1% in the over 60 s population who had been assessed almost a decade later, in 2015. This possible regional significant growth in numbers of cases with its associated increased burden of care is worrying. However, policy changes including public health messages, nutritional advice and using slightly different dementia assessments in the two studies may also reveal a stagnation in the percentage of people with dementia in this cohort in 2015, but this requires further investigation. Identifying predictors of dementia prevalence and possible protective and risk factors is crucial to help people maintain their independence, as there is no current dementia cure and its human and economic costs are high [1].

The Active Ageing Index (AAI) [7] was developed to illustrate the positive contributions older people can make to society. It was calculated across various countries in Europe, in South Korea and China to show geographical differences using just one index number consisting of combined weighted physical/material and psychosocial indicators to reflect the different aspects of the AAI. The AAI concept was criticized for not including variability associated with ageing, such as related to cultural differences, but also variability due to dependency, disability, ill health and need [8]. An updated Asian index was recently developed for Indonesia and Thailand, which incorporated several changes [9]. Many of the indicators of both indexes have been found to reduce dementia and dependency risk in previous epidemiological studies [3]. Of primary interest for the present study is which individual physical/material and psychosocial proxy AAI indicators contribute most to reduced dementia and dependency risk and which could be targeted for further policy change. We hypothesized (1) that dementia prevalence in the rural areas around Yogyakarta 2006/7 was similar to that found in the 2015 study when using the same assessments [6] (possibly because of policy changes) and (2) that Active Ageing components, in particular remaining healthy and engaging in physical and psychosocial activities would be associated with reduced dementia and dependency risk in this 2006/7 cohort. This study will act as a baseline study for a 15-year follow up of in the Indonesian cohort with data collected in the same geographical areas in 2021 to investigate whether more active age-friendly policies and increased dementia awareness and activities via Puskesmas (primary health care centers) in Indonesia have reduced the predicted increase in growth in dementia and dependency. Indonesia has been very proactive in developing age and dementia friendly national policies and laws in recent decades, including a pension scheme [10].

## 2. Materials and Methods

### 2.1. Participants

This Indonesian cross-sectional study included 705 eligible people who had to be 60 years of age or older at the time of testing, who had to reside in one of two rural health care districts (West Java, around Sumedang and Central Java, around Borobudur, near Yogyakarta) or an urban site (in Jakarta) and who had been visited between 2006 and 2007. The cohort and data used for this study have been described in more detail previously when investigating dementia prevalence in these different sites [5]. Briefly, prior to the study all community leaders and staff at local community health care centers or care institutes within the defined health care districts had been informed of the study and they subsequently forwarded this information to potential participants over 60. Participants had been asked to bring their carers if they had them, and to arrive in the morning at the local district health care centers (Puskesmas) at agreed dates if they were interested in participating. None of the inhabitants approached refused participation after they had been given information about the study by trained research assistants and all signed the informed consent forms before the start of the study. If a carer was present, they also signed the informed consent form. No incentive was offered.

The study was carried out between April and June 2006 in urban Jakarta and between December 2006 and February 2007 in the rural areas of Borobudur and Sumedang. Ethical approval (University of Indonesia, Jakarta; Loughborough University, UK R06/P21), governmental and local permits had been obtained prior to study-onset.

In West Java, at the rural Sumedang health care district site, all 203 Sundanese elderly 60 years of age and over who resided in the village of Citengah (a 2-hour drive from Bandung) were invited to come to the community health care center and were tested there (see questionnaires, measurements and cognitive tests used for assessment below) after they had given informed consent. At the Central Java site, covered by the Borobudur health care district (a 2-hour drive from Yogyakarta) all 214 Javanese elderly 60 years of age and over were included and were asked to come to the local Borobudur health care center to be tested after they had given informed consent. Those with limited mobility were visited at home (*n* = 2) if they had agreed to be visited there. For the urban areas in Jakarta (Covered by the Central, West and South Jakarta health care districts in Jakarta, in North-West Java), a sample of 288 elderly 60 years of age and over with mixed ethnicity (including 47% Javanese and 17% Sundanese, while other ethnic groups, such as Minangkabau, Chinese, etc., were less prevalent). The sample was included after giving informed consent. This ethnic distribution for Jakarta and the rural areas reflected the Indonesian census of 2005 as we wanted a representative Javanese sample of urban and rural different ethnic groups. Most of these participants (*n* = 164) were either attending the local community health care centers, lived in local care homes for older people covered by the health care district (*n* = 43 only in Jakarta), or were tested at home (*n* = 1) which was also covered by the health care district. Testing was done by fully trained and supervised research assistants between 8−11 a.m. to avoid circadian interference and the effects of heat.

### 2.2. Instruments, Questionnaires, Measures, and Assessment Procedure

After pilots and revisions of some contents, the translated forms of questionnaires and cognitive tests encountered no further problems and back-translation from Bahasa Indonesia (in Jakarta), Javanese (in rural areas around Borobudur) and Sundanese (in the Sumedang district) to English was done successfully for all tests. Participants were surveyed for demographic and other variables (such as health and lifestyle) using standardized questionnaires which were used as AAI proxy indicators (see Table 1). Answers were all substantiated by a carer when present (in about half of the suspected dementia cases and half of controls). The original AAI has 4 domains (see Figure 1 below, copied from open access [7]). We included proxy questions from our survey as indicators of the original AAI (see Table 1). Some of the AAI indicators required inclusion of data from participants 55−59 years of age but as our Borobudur site only included people over 60, we calculated and used proxy indicator frequency for the whole sample and districts only on the *n* = 705 who were 60 and over. Dementia risk before age 60 is also very low.

Outcome assessments were the following.

For this study the Instrumental Activities of Daily Living (IADL) questionnaire was implemented using a cut-off of ‘<9’ to assess dependency (see [5]). The questionnaire had been slightly adapted for rural populations (e.g., ‘using the telephone’ was changed to extending messages or using the telephone). The following cognitive screening tests were also measured. To assess memory function which is affected early in dementia, the adapted Hopkins Verbal Learning Test (HVLT) was used which is highly sensitive and specific to dementia [11]). This is a word learning test measuring episodic memory, which consists of 12 words from 3 low frequency categories (for version A: ‘human shelter’; ‘animals’ and ‘precious stones’). These words were all repeated 3 times to obtain a total immediate recall measure (‘learning ability’). Some items (of the precious stones category) were changed after a pilot study to adapt to local knowledge, creating a modified Indonesian version of this word list [5]. The Mini Mental Status Examination (MMSE [12]) is a gold standard dementia screening cognitive test and consists of a series of questions designed to measure change in cognitive status and to differentiate between normal age-related cognitive decline and the pathological cognitive decline that occurs in dementia. It was slightly adapted for local circumstances (e.g., seasons of the year were scored as wet or dry season, which was similar to the Hindi version used in India [13]).

An in-depth study was performed on the rural sample from Central Java to validate the cut-off scores of the cognitive tests and the IADL (alongside a carers report of cognitive impairment) to diagnose possible dementia [5]. For this study, Javanese Indonesian people over 60 years of age from 4 villages around Borobudur were asked to participate. Almost all 113 inhabitants agreed to participate and these volunteers were tested in a health care center by medical experts including psychiatrists and trained research assistants. The cognitive screening tests and IADL score cut-offs were validated against consensus based clinical dementia diagnoses from two expert psychiatrists, nurses and primary care physicians using a gold standard diagnostic instrument for dementia diagnoses from Cambridge (CAMDEX) and a computer-based diagnostic algorithm system developed and validated in Oxford using clinical data from over 200 post-mortem confirmed dementia cases and controls [14]. A similar validation study was done in a mixed ethnicity cohort in Jakarta. These studies confirmed the diagnostic accuracy of the cognitive tests and the algorithm used with the IADL cut-off score and carers report in the present study in identifying possible dementia [5]. For further self-report and carers’ confirmation of memory and other cognitive impairment and to identify progression of disease, questions based on the Dementia Questionnaire [15], were included, which were similar to the questions used in the 2015 study in the rural areas around Yogyakarta employing the AD8 questionnaire [6]. This allowed comparisons and investigation of our first hypothesis.

### 2.3. Statistical Analyses

To establish cognitive impairment/possible dementia in the present study, the combination of the two cognitive tests was employed, using the cut-offs on the total immediate recall of the HVLT (14.5) and the total score of the MMSE (24.5) to reflect cognitive impairment, and the IADL score (9) to reflect dependency [16] based on the validation studies [5,16]. Demographic characteristics per site are displayed in Table 2.

Descriptive analyses for AAI proxy indicators per site are shown in Table 3. Weighting and calculation of the AAI itself per site was not carried out as the exact indicators were not assessed and some variables (volunteering, poverty) had not been assessed in this survey. The indicator variables were all entered, then using stepwise backward logistic regression analyses to estimate the relative contribution of these indicators to risk of dependency (as established IADL < 9) and risk of dementia (cognitive impairment on both tests as indicated by cognitive test cut-offs and IADL < 9). For all analyses SPSS 22.0 was used with a *p*-value of <0.05 indicating significance. We included Holm-Bonferroni corrections by dividing *p*-values by numbers of tests (steps in the model) performed. For dementia this was 10 steps (corrected *p*-value stepwise = 0.005–0.006–0.006–0.007–0.008–0.01–0.012–0.017–0.025–0.05) and for dependency this was 11 steps (corrected *p*-values stepwise = *p* = 0.0045, 0.006–0.006–0.006–0.007–0.008–0.01–0.01–0.02–0.025–0.05).

## 3. Results

### 3.1. Demographics

Average age of the cohort was 69.4 (7.70) and 64.3% was female. Reflecting the census, in this sample, participants in Jakarta were on average younger and had higher levels of education than those tested in rural areas (Table 2). They were also more likely to be white collar workers, not live alone, but more likely to live in a care home or institute (15%), and less likely to own their own home, but to rent instead (9%).

### 3.2. Suspected Dementia and Dependency Prevalence among Sites

Participants from rural areas (all participants covered by the rural health centers near Borobudur and Sumedang, in Central and West Java, respectively) had a substantially higher risk of cognitive impairment/possible dementia using the cut-offs from Oxfordshire which had been validated in Indonesia. Around 43% of participants in both rural areas scored below cut-offs of both tests using the established algorithm, compared to the urban areas (with only 11% scoring below both cut-offs in Central and South Jakarta). Rural older people were thus 4 times more likely to score below cognitive test cut-off scores. Investigating IADL separately, as an indicator of dependency, also showed regional differences, with 1 in 5 of older people over 60 years of age in rural Borobudur and 1 in 10 in Sumedang and Jakarta experiencing dependency as assessed by IADL. Of all participants included, 3% in Jakarta, 7% in Sumedang and 16% in rural Borobudur scored below cut-offs for cognitive impairment and IADL (<9). Overall, in the total cohort suspected dementia prevalence in those over 60 years of age was around 9%. This was slightly higher than expected at 5%, based on data from India [12] which at that time had a similar population distribution. This was the case because of a very high percentage of possible dementia cases in the Borobudur district, near Yogyakarta in Central Java.

To test our first hypothesis and compare our 2006/2007 data against the 2015 data from these same rural areas close to Yogyakarta from a recent paper [6] using their MMSE cut-off of 21 and IADL > 1, rendered 30% of people with suspected dementia of the *n* = 214 over 60 assessed at that site in 2006/7. Adding the proxy questions from the Dementia Questionnaire for the AD8, with at least two self-reported cognitive impairments (memory, thinking, naming, orientation), showed confirmation in about 49–74% (depending on the question) of *n* = 65 people suspected of dementia, suggesting that around 15–22% of this population had dementia in 2006/7 using the algorithm used in the 2015 paper. The 2006/7 data thus provided a similar range to those reported in the same area in 2015 with around 20% of people over 60 suspected of having dementia [6]. However, surprisingly, carers were much less likely to confirm the self-reported cognitive impairment (only 12–23% of *n* = 65) which would result in only 4–7% of people in the rural Borobudur area possibly having dementia. Whether this is due to respect of carers and under-representation of actual issues or inaccurate overrepresentation of people with dementia is unclear.

In the whole cohort, about half of carers stated there were no memory problems in the elderly who scored below the cut-offs on the HVLT and MMSE. Overall, data of *n* = 136 carers indicated that around *n* = 40 (5.6% of the total cohort) of suspected cases were reported to have memory problems, of whom *n* = 22 had gotten progressively worse over time and who may have had Alzheimer’s disease (55% of all suspected dementias). Taking these data together indicated that 3–7% of people, who were included in our sample and who were over 60 years of age could be afflicted with dementia in 2006/2007, but as discussed above; there was large inter-regional variability.

The AAI includes positive (physical/material as well as psychosocial) indicators in older people and their communities and these indicators (see Table 3 for the frequency distributions of the individual variables for the over 60 s per district) have previously been associated with a lower risk of dementia and dependency [3]. Table 3 shows large differences between sites which could impact on outcome variables as predicted by our second hypothesis.

Overall, risk for dementia was almost double in rural areas (2-3x between districts). However, using stepwise backward logistic regression to identify the main individual indicators for dementia risk, educational differences between districts possibly largely explained these differences (OR = 0.62, 95% CI = 0.51–0.77, *p* < 0.0001). Dementia risk was independently associated with an older age (OR = 1.06, 95% CI = 1.04–1.09, *p* < 0.0001). Reading frequently (OR = 0.76, 95% CI = 0.62–0.93, *p* = 0.008) and attending community activities (OR = 0.76, 95% CI = 0.62–0.93, *p* = 0.007) reduced dementia risk even after Holm-Bonferroni corrections. Some variables remained in the model, but were not significant after Holm-Bonferroni correction, including doing sports (OR = 1.86, 95% CI = 1.02–3.37, *p* = 0.04, ns after Holm Bonferroni correction), having poor physical health (OR = 1.27, 95% CI = 0.99–1.64, *p* = 0.06, ns after Holm Bonferroni correction) and attending social gatherings (OR = 0.84, 95% CI = 0.69–1.01, *p* = 0.07, ns after Holm Bonferroni correction), in reducing risk for dementia in the model. Other aspects of socioeconomic status (house ownership, occupation), attending a doctor, living circumstances, gender, feeling secure, being more or less happy or more or less physically active than others, watching TV or talking to friends were all not significantly associated with cognitive impairment/possible dementia in these analyses.

For the IADL/dependency by itself as an outcome, similar results were seen. Risk for dependency was associated with an older age (OR = 1.12, 95% CI = 1.08–1.17, *p* < 0.0001) and residing in more supportive living circumstances (OR = 1.54, 95% CI = 1.22–1.93, *p* < 0.0001). Poor health increased dependency risk almost 3-fold (OR = 2.96, 95% CI = 2.00–4.33, *p* < 0.0001). Independently, engaging in sports (OR = 3.27, 95% CI = 1.36–7.81, *p* = 0.008, reverse scoring) and attending community activities (OR = 0.54, 95% CI = 0.37–0.77, *p* = 0.001) were associated with reduced risk for dependency, which remained significant after Holm-Bonferroni correction. However, reading (OR = 0.74, 95% CI = 0.56–0.97, *p* = 0.02, ns after Holm-Bonferroni correction) and going to social gatherings (OR = 0.76, 95% CI = 0.59–0.99, *p* = 0.04, ns after Holm-Bonferroni correction) remained in the model, but after correction no longer significantly reduced risk of dependency. Other aspects of socioeconomic status (site, education, occupation and house ownership), attending a doctor, gender, feeling secure or happy, watching TV or talking to friends were not entered in the model or associated with dependency in these analyses.

## 4. Discussion

The overall estimate over 3 multi-ethnic sites on Java in this 2006/7 study was that around 9% of people screened over 60 years of age were suspected of having dementia. There were large interregional differences (3–16%). Risk for both dementia and dependency was highest in Borobudur (for dependency 1:5, as compared to 1:10 of people over 60 in the other two areas). Participants here were of an older average age (but this was only of an average of 2 years more). Gender, ethnicity or site did not contribute to the models for dementia, indicating no need for adjusted cut-offs or gender/ethnicity stratified analyses. However, education possibly explained the differences in dementia prevalence in urban and rural districts. In the Borobudur district almost half of people over 60 had not obtained formal education. Our earlier validation paper suggested that different cut-offs for age, education, ethnicity and gender were not needed for dementia screening in this area [5]. However, a recent paper of rural data collected in the same area almost a decade later reported dementia was also predicted by a lack of education and an older age.

In the 2006/7 analyses, we had used an additional memory recall test (the HVLT) to assess dementia risk, which was not sensitive to education in our earlier analyses, whereas the MMSE used in the more recent study [6] and also in our earlier study [5] was affected by educational levels, especially affecting performance in dementia cases. This could have led to finding a slightly higher dementia prevalence in that same district in 2015 when only using the MMSE, but as seen, much depends on which questions are asked to whom to verify cognitive impairment impacting on activities of daily life. In 2006/7 other indicators of socioeconomic status such as occupation, home ownership, attending a doctor or feeling secure were not significant predictors of care needs in the overall model. In the 2015 analyses of this area, authors reported that apart from an older age and lower education, the female gender, unemployment and a history of stroke also contributed independently to dementia risk [6]. In contrast, in 2006/7, we found in the whole cohort that men had a slightly—but non-significant -increased risk of dementia in all sites and that there was significantly more dependency in older men (21% vs. 17% of women) in rural Borobudur in 2006/7. In our analyses, there were no substantial gender differences with regard to occupation, as many older women in rural Borobudur still worked in 2006/7.

Most older people in rural parts still actively contribute to family finances [17] by often doing physically demanding manual labor. Occupation did not contribute to dementia or dependency risk in our overall analyses. However, exploratory analyses of the Sumedang district suggested increased risks of dependency in manual laborers. Self-report health in this district was also on average lower than that in the other two areas, while being physically active was reported to be higher. It could be that these physical demands (and poor nutrition?) exceeded physiological capacity, leading to lower health associated with dependency. Overall, we found that living independently, a good physical health status and engaging in health promoting psychosocial activities- such as sports (mainly in Jakarta) and attending community activities in all areas- were associated with higher levels of independence. For dementia, reading (mainly in Jakarta in exploratory post hoc analyses) and attending community activities in all areas was associated with reduced risk.

It could be proposed that these ‘risk’ and ‘protective’ factors are not causally related to- but rather a consequence of- increased mental and physical dependency, such as needing supportive living circumstances, but possibly also not being able to read or do sports (anymore) or attend community or other social activities. However, our previous observational work showed that long term engagement in physical activities before- but also after- dementia onset reduced dependency due to memory impairment [18]. Not engaging with intellectually stimulating activities such as reading, using computers or other information technology [19], experiencing loneliness, not being married and having a lack of quality social contacts [20] all increased dementia risk years before dementia onset in a UK cohort. Our randomized controlled trials with physical exercise classes reduced physical dependence and biological risk factors in institutionalized prefrail and frail older women [21] and improved memory in those with and without dementia [22].

These and other studies suggest that engaging in these physical and psychosocial activities can contribute to reduced risk for dementia at all stages of prevention [3]. Our earlier work showed that the better educated, more affluent and slightly younger Jakarta based cohort was more likely to eat healthier diets and exercise [23]. This cluster of behaviors could be related to their lower dementia risk [3]. Engaging in sport and reading were mainly protective in urban Jakarta as exploratory post hoc analyses stratified for district suggested. For instance, while 63% of the older people in Jakarta and 47% of people in Sumedang did gymnastics, none of the interviewed in Borobudur engaged with this type of exercise. Attending community activities were also very important in all areas and age-friendly community design with communal areas, parks and Puskesmas, with more frequent social and community activities, including knowledge transfer of dementia symptoms and interactive games which include sport or exercise can be important facilitators of this. To support reading, a focus on early education and more library facilities in rural areas with a further focus on reducing illiteracy should be a focus of policy. Indonesia is very proactive in dementia prevention with public health messages on good diets containing tempe and fruits, smoking cessation, and maintaining physical activity and many Puskesmas we visited already showed good practice. Training is done by Universities, such as Universitas Indonesia and Respati University for the ‘posbindu caders’ volunteer community health workers who share this knowledge on older people’s nutritional needs, and perform checks on independence and dementia screening in community health centers or Puskesmas (https://scholarhub.ui.ac.id/ajce/vol4/iss2/12/ (accessed on 30 June 2021)).

Good perceived health was an independent and important factor to reduce dependency and to some extent dementia risk. For policy guidance unplanned analyses exploratory analyses showed that in 2006/7, access to health care was not different between urban and rural areas (health centre visits or outpatient hospital visits were reported by between 42–46% of the Jakarta and Borobudur sample, to 61% in Sumedang). On the other hand, attending or consulting a physician (reported by 10%, respectively 36% of older people in rural Borobudur, which was between 2–4× lower than in the other areas, where more than half had used these resources), as well as medication use (reported by 20% in Borobudur, with around 53–58% in the other areas) and admission to hospital (reported by 8% in Borobudur, with 11–17% in the other areas) were all lower in Borobudur. This access and use of health care will be further investigated in our 2021 survey to evaluate past and promote future policy focus on supporting these older people by providing better access to these resources locally. Most Puskesmas in 2006/7 had limited health care checks, once or twice a month only.

There are several limitations to the current study. Firstly, there was large variation between the sites and demographic differences in the distribution of age, ethnic differences, occupations and educational attainments between rural and urban samples may have acted as systematic confounds and are difficult to control for. We could not investigate the actual AAI in different districts for a better comparison with EU and other Asian countries, as we had insufficient proxy indicators. However, some of the indicators may lead to errors in interpretation of the role of the AAI in preventing poor health and dependency. For instance, house ownership was common in rural poor areas, whereas renting was more prevalent, but also very expensive in Jakarta. In addition, only those who could afford not to work, no longer had a job or instead mainly did white collar, less physically demanding work should be contrasted to those in the rural districts where the majority were blue collar workers and usually very poor—which both may have affected their health and risk for dependency, due to wear and tear (e.g., osteoarthritis, which is common in Indonesia) and a lack of access to medical resources to treat disabling disease.

Other morbidity contributions to dementia including stroke were not assessed in 2006/7 to also investigate secondary potentially reversable or treatable dementias which could explain its weak association with poor reported health in our 2006/7 analyses. Carer’s report of cognitive problems was also lower than expected from the test results. The more recent study in rural Borobudur suggested that carers had poor knowledge of dementia [6] Alternatively, cognitive tests such as the MMSE perhaps do not reflect everyday problems, or carers give a lower negative report out of respect for the elderly. In two villages visited, people reported that ‘pikun’ (dementia) did not exist. However, when we asked questions whether someone in the village was suffered from planning or memory dysfunction, recognizing people or getting lost, a similar dementia prevalence was found as in European countries (around 5%). Depression was not assessed formally and could not be taken along in analyses and this needs further exploration. We want to investigate which factors improve mental health, a known risk factor for dementia. In exploratory analyses happiness was correlated with feeling secure, but also playing more sports, walking more and being socially active, some of which (playing sports, being socially active) were entered in the model. To better aid public health interventions and health policy, more focused research needs to be conducted, especially in the role of close carers to prevent increasing care needs (for instance, in Sumedang living circumstances did not predict dependency, but here more older people were still living with a spouse, which could have off-set needs). Most important is to ask people themselves what they feel works in their community to keep older people happy, active and independent and our 2020/2021 study is doing this.

In Indonesia, at the time of testing, average life expectancy for men was 65 years and for women it was 69 years. People may have thus not survived to be old enough to be at risk for dementia, leaving only healthy survivors (who have not died of heart disease or stroke, etc.) to obtain an older age, who then have a later risk for dementia [3]. While our data suggested that this may the case in Jakarta, with fewer than expected cases in the older age strata, rural areas showed more suspected cases based on non-educated adjusted cognitive cut-offs with increasing age, partly refuting these hypotheses. Alternatively, mainly the affluent and well educated can afford to retire in Jakarta, which is expensive. With this comes a cluster of health promoting lifestyles such as better diet, sports, being affluent, feeling more secure, and having better access to physicians, medicine and hospitals. Older less educated and poorer people either migrate back to rural Borobudur or stay in Jakarta, but either way are at higher risk -due to access to fewer (medical) resources-, which could help reduce risk of reversible dementias and poor health, leading to dependency. The extent to which the characteristics and type of manual labor possibly leading to poor health in Sumedang may be associated with increased dependency also requires more research. At the time only around 5% of people were estimated to have a pension and in recent years the government has put much effort into setting up a pension scheme. Whether this policy has reduced the associations found in our 2006/7 study remains to be seen in our planned analyses on the 2021 data.

For future research, as we are interested in reciprocal care [24], we will also include other variables from epidemiological research on other known risk and protective factors shown to contribute to a reduced risk of dementia and compression of needs in older people, including diet and morbidity. We would ultimately like to assess associations of the individual and local available resources or indicators with morbidity and biomarkers known to affect cognitive disability, dependency and dementia risk over time to inform screening programs and policy makers. For this purpose (see Figure 2), we have rearranged some of the original indicators of the AAI and renamed these under physical/material vs. psychosocial resources. We also added additional risk and protective morbidity factors associated with dementia and dependency needs.

This new Compression of Needs model brings the probability of the outcome (risk for increased needs due to dementia) partly back to the individual, who can make personalised choices on how to reduce their individual risk and assess which changes they can make to reduce this risk. In addition, by including formal and informal care and support resources to psychosocial indicators possible identified, off-setting through these factors can inform policy makers where to direct resources (e.g., in rural areas where carers/family have left, more formal primary and home care provisions and better access to medicine and short term stay in hospitals could take their place). While dementia prevalence was low in Jakarta in 2006/7, it was higher in rural areas. On the one hand, this morbidity could perhaps have less impact in the rural settings, which require less complicated everyday decision making. On the other hand, with increasing Westernization, and more young people moving away from rural parts, dementia may be an increasing issue for isolated elderly who remain behind with little support. This support works both ways, as older people still contribute to family finances [17] and dementia means loss of economic contribution of the person affected, but also of their carers, who need to give 24-hour care in the late stages of the disease.

With our estimates of almost 2 million older people affected by dementia by 2025, this means that at least 4 million of people over 60 years of age and/or 2 million of their children, who could still substantially contribute to the Indonesian economy, can then no longer do so. Targeted public health interventions and support could possibly offset this risk in individuals and communities.

## Figures and Tables

**Figure 1 ijerph-18-08235-f001:**
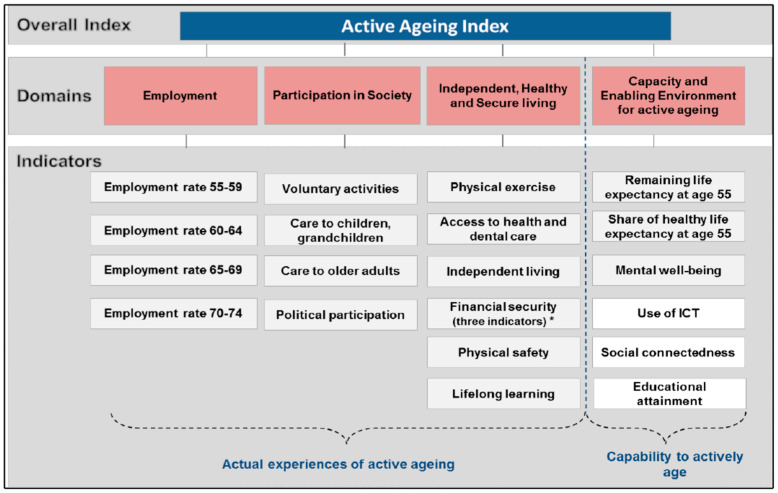
The domains and the indicators of aggregated Index, AAI.

**Figure 2 ijerph-18-08235-f002:**
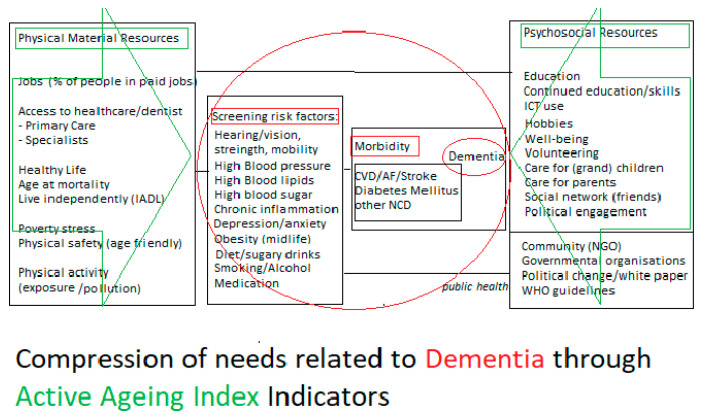
The Compression of Needs Model (Hogervorst, first discussed ADI, 2020 Singapore).

**Table 1 ijerph-18-08235-t001:** Questions from the 2006/7 survey to approximate AAI indicators in different sites in Indonesia.

For domains and indicators, the following proxy questions from the survey were used:
**1. Employment** - ‘How old are you’ (what was your age at your last birthday in years)?—used to rank indicators in AAI. - ‘What is your job or profession?’ (1 = not working/retired, manual work (e.g., labourer) had the highest score).
**2. Participation in society/social participation** - Voluntary and social activities: ‘Do you participate in social gatherings/engage in community activities’. - Care for children AND older people (we had no data available, but we used data from living arrangements).
**3. Independent, healthy and secure living** - ‘Compared to people of you own age, how do you rate your physical activity?’ (lower score means less, higher score means being more physically active than others). - ‘Do you play sport regularly?’ (yes = 1, no = 2). - ‘Do you attend your doctor or health provider?’ (yes = 1, no = 2) which was used for access to health care or ‘the ability to get medical treatment’ in the Asian AAI and for ‘access to health care and dentistry’ in the AAI. - ‘With whom do you live at this moment?’ (a low score reflects more independent living, a higher score reflects more supported living, with others or in a care home/institution). We could not use ADL/IADL as an indicator for dependency (as is done in the Asian AAI) as it was our outcome variable. - ‘House ownership’ (own house 1, rented = 2, with others = 3, institution = 4) as a proxy of financial security. - This was also used in the Asian AAI. We had no data on median income of the areas in 2006/2007. - ‘Do you feel secure?’ (1 = more, 2 = same, 3 = less than others) This was moved to domain 4. in the Asian AAI. - ‘Do you read ?’ (used as a proxy of continuing education, a higher score is ‘(very) regularly’, on a 5-point scale.
**4. Capacity and enabling environments** - Remaining life expectancy at age 55 (we used oldest age reported in the district, as all inhabitants of the district participated in the survey and age was entered continuously in regression analyses). - Remaining healthy life (including health data). - ‘In general, how would you say your health is’ (a higher score indicates poor health, on a 5-point scale). - Subjective well-being was assessed by asking ‘Are you happy?’ (1 = yes, 2 = no). - We did not have data on the CES-D or WHO-5 index for mental well being and subjective well being. - ‘Do you watch TV’? (was used a proxy of ICT use, as most people did not have internet access in 2006 in rural parts), a higher score is ‘watching TV (very) regularly’ on a 5-point scale. - Social connectedness: ‘Do you talk to friends, neighbours or family ?’ (a higher score is (very) regularly on 5-point scale), which is a similar questionnaire as is used in the Asian AAI - -‘What was the highest education level you graduated from?’ (for educational attainment, with a higher score indicating a higher level of education, from high school onwards).

Using the total model with all variables included rendered fewer than 10% missing data for the whole cohort.

**Table 2 ijerph-18-08235-t002:** Demographics in the different sites in participants over 60 years of age. Bold indicates differences by site (data given in percentages or means, SD = standard deviation, r = range).

Variable	Jakarta	Sumedang	Borobudur
*Age*	68.6 (SD = 7.5, r = 60–90)	69.4 (SD = 7.5 r = 60–98)	**71.3** (SD = 7.9 r = 60–90)
% women	71%	61%	58%
*Education*			
None	13%	19%	**46**%
< Primary school	**18**%	54%	77%
High School l>	**46**%	2%	3%
*Occupation*			
No job	**62**%	13%	8%
White collar	**37.5**%	15%	20%
Blue collar	**5.5**%	72%	72%
(farmer, laborer, fisherman)			
*Own house*	**68**%	92%	95%
Rent	9%		
Other	24%	7%	5%
*Living circumstances*			
Alone	**4**%	16%	13%
With spouse	15%	**28**%	13%
+ child	44%	36%	**13**%
Family	22%	20%	**61**%
Care home	**15**%	0%	0%

**Table 3 ijerph-18-08235-t003:** Proxy indicators of the Active Ageing Index and outcomes over different sites/districts. Bold indicate differences between districts.

Variables	Jakarta	Sumedang	Borobudur
*N*	288	203	214
Dementia	**3%**	7%	**16%**
IADL < 9	10%	11%	**19%**
**AAI Indicators**			
*A. Are you currently employed*			
Not working	11%	13%	**8%**
Blue collar (farmer/labourer)	**2%**	72%	72%
*B. Unpaid contributions to/within society*			
1. do you volunteer via an organisation- Do you engage in community activities?	**30%**	18%	19%
2. do you take care of your (grand)children- Do you live with your children?	62%	51%	64%
3. do you take care of older people- Do you live with a spouse (without child)?	15%	**28%**	13%
4. do you engage in political activities- Do you go to social gatherings?	15%	16%	13%
*C. Live independently, healthy and safe/secure*			
1. Are you physically active/exercising almost every day- Do you play sports (yes/no)?	**80%**	33%	**3%**
Are you physically active regularly/more or less than others?	10% 41%	**21% 40%**	10% 18%
2. do you have access to health services (and dentist)- Do you attend a health care provider?	54%	48%	**10%**
3. Do you live independently, i.e., single/couple?	19%	**44%**	26%
4a. Are you financially secure—home ownership?	**68%**	92%	95%
4b. Are you at poverty risk = below threshold-no data	Affluent		
5. Is it physically safe for you to walk in the dark locally- Do you feel more or less secure?	46% 2%	36% 6%	**6% 3%**
6. Have you received any education/training in last 4 weeks- Do you read? (often/very often)	**60%**	27%	12%
*D. Capacity enabling environment to age actively*			
1. Remaining life expectancy-age oldest participant	90	**98**	90
2. and which proportion of this is healthy- In general how would you say your health is (good-excellent)?	70%	**51%**	80%
3. Mental well-being- Are you more or less happy than others you know?	48% 2%	40% 10%	**9% 6%**
4. Use ICT >week- Do you watch TV often/v often?	**75%**	53%	43%
5. Social connectedness who meets friends/colleagues/relatives- Do you talk to friends, neighbours family often/very often?	68%	55%	43%
6. Educational attainment- (% with upper secondary or tertiary education (high school>).	**46%**	2%	3%
No formal education at all (This category was included in Table 2 under < primary school (for people who did not finish primary school OR had no education) but is here displayed by itself in Table 3).	13%	19%	**46%**

## Data Availability

Data are available upon request from the author.

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
