# Peer review of "Dementia and Dependency vs. Proxy Indicators of the Active Ageing Index in Indonesia"

_ijerph, 2021, doi:10.3390/ijerph18168235_

Round 1

Reviewer 1 Report

This is a well done study that will serve as background  for future research in in the important area of dementia prevention and treatment.zzz few typos need correcting-lines 36, 49, 61, 306.What is lederhosen line 364? The various colors on my copy used for emphasis need better explaining.

g

Author Response

  • Thank you for your kind feed-back. We have addressed the comments as follow
  • Few typos need correcting-lines 36, 49, 61, 306.What is lederhosen line 364?
  • A: We have corrected the typos but could not find a reference to Lederhosen (which are short leather trousers worn by Germans)
  • The various colors on my copy used for emphasis need better explaining.
  • A: In table 2 and In table 3 the bold, respectively highlights indicated the differences between districts which has been now added

Reviewer 2 Report

The authors provide an interesting view on the topic. It has good supporting evidence and it is of interest for the scientific community. Please take in consideration the comments provided. 

The paper has been reviewed. It is acceptable but the authors should consider the suggestions provided.

Abstract

L12: The citation which matches with this statement, in the Introduction section ([1]), refers to Indonesia instead of worldwide. 

Introduction

L44: Interesting point, care to elaborate more or provide references

L65: Please, specify more in-depth the objectives of the study, according to your hypothesis.

L70: Authors meant scheme. 

Materials and Method

L71: Materials and Method section would benefit from a more precise separation in subsection, i.e. Participants, Instruments/Questionnaires/Measures, Assessment Procedure and Statistical Analyses. This helps the reader to find information regarding specific questions of the study.

L72: Please, include eligibility criteria (inclusion and exclusion) for participating in the study.

L73: Please, even though sample is described in [5], include the sex proportion and age mean and standard deviation for the complete sample. Correct extra spacing after 705.

L88: Please, provide information about the ethical committee approval number. 

L91: Were tested for what? 

L157: Why forward procedure?

L160: Have you included any p-value correction for multiple comparisons (for example, Bonferroni)? If this is not the case,  any method has to be applied in order to correct this issue

Results

L216-229: This paragraph should be placed in the Discussion section.

L251: A p-value is not a trend. Please, correct this and include all the information for the non-significant results (OR, CI).

L256: You should also indicate whether the results remain significant after multiple comparisons correction.

L257: If you do not consider this analysis to have enough power, it should be removed.

L258: If you maintain analyses per site, statistical results must be included, not only a statement pointing to the significant variables. 

L273: Same as for L258. All results you report must include the numerical outcomes (statistic, odds ratio in this case, confidence interval, p value).

L276: Same as for L258 and L273.

L283-289: Please, indicate the percentages of Table 2 or 3 you are referring to in this paragraph. 

Discussion

L230: Any reference for this statement?

Additional information

L474: Please, include the reference number or numbers of your Ethics Committee approval.

L475: Where is the informed consent statement?

L477: How is the data going to be available in order to improve replicability and open science?

Author Response

Thank you for your kind words and review. We have revised the paper as follows:

Abstract

L12: The citation which matches with this statement, in the Introduction section ([1]), refers to Indonesia instead of worldwide. 

A: The website the citation was taken from cites: 'Globally, Indonesia has the fifth-largest elderly population in the world'

Introduction

L44: Interesting point, care to elaborate more or provide references

A: this study is subsequently described in the next sentences as follows

'This study showed large differences in dementia prevalence in those over 60 years of age between urban (3%) and rural sites (7-16%), with the highest rate in rural areas close to Yogyakarta, which also had the oldest and least educated population [5]. A recent paper [6] suggested even higher rates of dementia prevalence in this area, of 20.1% in the over 60s population who had been assessed almost a decade later, in 2015. This possible regional significant growth in numbers of cases with its associated increased burden of care is worrying. However, policy changes including public health messages, nutritional advice and using slightly different dementia assessments in the two studies may also reveal a stagnation in the percentage of people with dementia in this cohort in 2015, but this requires further investigation.'

L65: Please, specify more in-depth the objectives of the study, according to your hypothesis.

A: We have added in row 65 

'We hypothesized 1) that dementia prevalence in the rural areas around Yogyakarta 2006/7 was similar to that found in the 2015 study when using the same assessments [6] (possibly because of policy changes) and 2) that Active Ageing components, in particular remaining healthy and engaging in physical and psychosocial activities would be associated with reduced dementia and dependency in this 2006/7 cohort. '   

L70: Authors meant scheme. 

A: This is revised, thank you

Materials and Method

L71: Materials and Method section would benefit from a more precise separation in subsection, i.e. Participants, Instruments/Questionnaires/Measures, Assessment Procedure and Statistical Analyses. This helps the reader to find information regarding specific questions of the study.

A; This has now been altered in the text

L72: Please, include eligibility criteria (inclusion and exclusion) for participating in the study.

A; This has now been added to the methods as follows

'This Indonesian cross-sectional study included 705 eligible people who had to be 60 years of age or older at the time of testing, who had to reside in one of two rural  health care districts (West Java, around Sumedang and Central Java, around Borobudur, near Yogyakarta) or an urban site (Jakarta) and who had been visited between 2006 and 2007. '

L73: Please, even though sample is described in [5], include the sex proportion and age mean and standard deviation for the complete sample. Correct extra spacing after 705.

A: Average age of the cohort was 69. 4 (7.70) and 64.3% was female. This was added to the results section. Spaces have been removed.

L88: Please, provide information about the ethical committee approval number. 

A; This has now been added to the methods and at the end of the manuscript

L91: Were tested for what? 

A: We added '...were tested (see questionnaires, measurements and cognitive tests) 

L157: Why forward procedure?

A: stepwise backward procedure was used instead as we wanted to include only those variables that gave a significant contribution to the model. 

L160: Have you included any p-value correction for multiple comparisons (for example, Bonferroni)? If this is not the case,  any method has to be applied in order to correct this issue

A; We have removed all exploratory analyses per district to reduce chance findings. We used the stepwise backward method for logistic regression (rather foward entry and rather than individual bivariate models for each of the variables) with all indicators of interest entered and then removed stepwise  to exclude multiple separate analyses so only variables that were significant were entered in the final model. We have included Holm Bonferroni corrections

Results

L216-229: This paragraph should be placed in the Discussion section.

A: We have now made it clearer in the introduction that this analyses to compare our 2006/7 study to the 2015 study was part of our hypotheses 

L251: A p-value is not a trend. Please, correct this and include all the information for the non-significant results (OR, CI)

We have changed this.

L256: You should also indicate whether the results remain significant after multiple comparisons correction.

A; we have done this

L257: If you do not consider this analysis to have enough power, it should be removed.

We have removed these sections

L258: If you maintain analyses per site, statistical results must be included, not only a statement pointing to the significant variables. 

L273: Same as for L258. All results you report must include the numerical outcomes (statistic, odds ratio in this case, confidence interval, p value).

L276: Same as for L258 and L273.

we have done this

L283-289: Please, indicate the percentages of Table 2 or 3 you are referring to in this paragraph. 

These sections have been removed

Discussion

L230: Any reference for this statement?

This section has been removed

Reviewer 3 Report

This is a cross sectionnal study conducted in 2006 which describes some associations between sociodemographics variables and cognitive impairement and dependency in Indonesian people > 60 years.

Main points:

The participants are not representative of the Indonesian population and it is not clear how they were selected. This could introduce a significant bias. 

The study is quite old, why publish it now? The authors cite numerous published materials on this study. Does the current manuscript contain duplicate data or previously published content?

The authors do not mention participants who are illiterate. How many are there? Is their proportion greater in the rural district? This point is important because it may influence the results of cognitive tests independently of the presence of dementia.
The authors talk about dementia, but they did not use appropriate methods to diagnose this condition. As they used screening tests, they should talk about cognitive impairment and not dementia.
The authors show associations between certain socio-demographic variables and cognitive impairment or dependency. The interpretation of these results is misguided: the authors repeatedly state that such and such a variable increases or reduces the risk of dementia or dependency. Interpretation of such associations should be more cautious as it is sometimes cognitive impairment or dependence that may influence the independent variables, and there may be confounding factors.
The authors indicate that they are conducting a longitudinal study to see how this cohort develops. It would be preferable to wait for the results of this longitudinal study to investigate more carefully the risk factors for incident dementia or cognitive impairment in this population.

Minor points: Many typing errors. Why are some cells in the Table 3 coloured yellow? The way some of the variables in Table 1 are scored is not very explicit. What does "Not fin. prim" mean? and "lansia". To what extent are the cognitive tests used in the study validated in the Indonesian population? Certainly the back-translation process is not sufficient due to the large cultural differences. This should be discussed in depth.

Author Response

Thank you for the review of the paper, we believe it has substantially improved the paper. See below our comments

Main points:

The participants are not representative of the Indonesian population and it is not clear how they were selected. This could introduce a significant bias. 

A: selection was done based on the health care districts to obtain urban Javanese older people (the mix of ethnicities reflected the 2005 census). We also selected two rural samples from Sundanese and Javanese people which are the most common ethnicities on Java which were also covered by well defined health care districts as described in the methods  

The study is quite old, why publish it now? The authors cite numerous published materials on this study. Does the current manuscript contain duplicate data or previously published content?

A: While we published before on dementia prevalence in these districts, this study is done using new variables within the Active Ageing framework which is novel. This analyses will then be performed on the same districts using follow-up data from 2015 and 2021 to investigate trends over time

The authors do not mention participants who are illiterate. How many are there? Is their proportion greater in the rural district? This point is important because it may influence the results of cognitive tests independently of the presence of dementia.

A:we had no data on illiteracy but we had data on education which was low in rural areas. As can be seen in the revised analyses now education substituted districts.

The authors talk about dementia, but they did not use appropriate methods to diagnose this condition. As they used screening tests, they should talk about cognitive impairment and not dementia.

A: As outlined in the introduction and methods, extensive validation studies were performed on the use of the algorithm to diagnose dementia. The same algorithm had been used in Oxfordshire memory clinics to screen for dementia. We have altered the term dementia to possible dementia as we did not have data on co-morbidity which could have caused the cognitive impairment

The authors show associations between certain socio-demographic variables and cognitive impairment or dependency. The interpretation of these results is misguided: the authors repeatedly state that such and such a variable increases or reduces the risk of dementia or dependency. Interpretation of such associations should be more cautious as it is sometimes cognitive impairment or dependence that may influence the independent variables, and there may be confounding factors.

A: We have removed the sections on individual districts which included the comments by SES and also acknowledged the possible confounding of direction of effects found. 

The authors indicate that they are conducting a longitudinal study to see how this cohort develops. It would be preferable to wait for the results of this longitudinal study to investigate more carefully the risk factors for incident dementia or cognitive impairment in this population.

A: We would like to submit this paper with its novel analytical approach as including all studies (baseline and 2 follow-ups) would result in a very large paper which would be hard to understand 

Minor points: Many typing errors. Why are some cells in the Table 3 coloured yellow?

A: we have removed the typing errors

A: we have added explanations on bold and highlights in table 2 and 3

The way some of the variables in Table 1 are scored is not very explicit. What does "Not fin. prim" mean? and "lansia".

A: not fin prim was explained in the legend below the table but has been changed to < primary school (it was 'not finished primary school) and lansia was explained in the introduction  meaning old people 

To what extent are the cognitive tests used in the study validated in the Indonesian population? Certainly the back-translation process is not sufficient due to the large cultural differences. This should be discussed in depth.

A: the introduction and methods describe the validation studies which had been conducted and showed high sensitivity and specificity for dementia (>93%). We also added these comments in the discussion

Thank you kindly for your comments

Thank you kindly